# Action-Constrained Imitation Learning

Chia-Han Yeh [* 1]   Tse-Sheng Nan [* 2]   Risto Vuorio [† 3]   Wei Hung [1]   Hung-Yen Wu [1]   Shao-Hua Sun [4]
Ping-Chun Hsieh [1]

## Abstract

Policy learning under action constraints plays a central role in ensuring safe behaviors in various robot control and resource allocation applications. In this paper, we study a new problem setting termed Action-Constrained Imitation Learning (ACIL), where an action-constrained imitator aims to learn from a demonstrative expert with larger action space. The fundamental challenge of ACIL lies in the unavoidable mismatch of occupancy measure between the expert and the imitator caused by the action constraints. We tackle this mismatch through *trajectory alignment* and propose DTWIL, which replaces the original expert demonstrations with a surrogate dataset that follows similar state trajectories while adhering to the action constraints. Specifically, we recast trajectory alignment as a planning problem and solve it via Model Predictive Control, which aligns the surrogate trajectories with the expert trajectories based on the Dynamic Time Warping (DTW) distance. Through extensive experiments, we demonstrate that learning from the dataset generated by DTWIL significantly enhances performance across multiple robot control tasks and outperforms various benchmark imitation learning algorithms in terms of sample efficiency. Our code is publicly available at https://github.com/NYCU-RL-Bandits-Lab/ACRL-Baselines.

## 1. Introduction

Reinforcement learning (RL) finds policies that maximize cumulative rewards through interactions with environments. However, in many real-world applications, interacting with environments can be costly or dangerous, and designing an effective reward function that consistently encourages the desired behavior in all situations could pose a significant challenge. In such cases, imitation learning (IL) (Pomerleau & A, 1991; Ho & Ermon, 2016) offers a compelling alternative. Rather than learning from a reward function through trial and error, IL learns a policy directly from a set of pre-collected expert demonstrations, which are transition data logged from a near-optimal policy.

In many real-world tasks, imposing constraints that define *feasible sets of actions* to ensure the safe and proper functioning of agents is necessary. Classic examples include optimal allocation of network resources under capacity constraints (Xu et al., 2018; Gu et al., 2019; Zhang et al., 2020) and robot control under kinematic limitations that prevent damage to the physical structure of robots (Pham et al., 2018; Gu et al., 2017; Jaillet & Porta, 2012; Tsounis et al., 2020). While there has been substantial research on action-constrained reinforcement learning (ACRL) (Kasaura et al., 2023; Lin et al., 2021; Brahmanage et al., 2023; Chen et al., 2024; Hung et al., 2025), surprisingly, little attention has been given to its imitation learning counterpart. Hence, in this work, we propose a novel problem, Action-Constrained Imitation Learning (ACIL), which concerns learning an agent under action constraints from a constraint-free expert demonstration set, *e.g.*, learning to control a robot arm with a limited power supply from trajectories collected from a more powerful robot arm (*i.e.*, higher torque limit).

Can we adopt existing ACRL methods to address the ACIL problem? To ensure that the actions generated by the policy adhere to specific constraints during both training and evaluation, most existing ACRL methods incorporate a projection layer on top of the policy network (Chow et al., 2018; Liu et al., 2020; Gu et al., 2017). However, such an approach can cause problems in IL. Most IL approaches aim to minimize the discrepancy between the occupancy measure of the expert demonstrations and that of the imitator (Pomerleau & A, 1991; Ho & Ermon, 2016). When expert actions are outside the feasible action set, the projection layer can prevent the imitator from accurately matching the occupancy measure of the expert, especially in cases with more restrictive action

---

[†]Work conducted while at the University of Oxford.

[*]Equal contribution  [1]National Yang Ming Chiao Tung University, Hsinchu, Taiwan [2]University of Illinois at Urbana-Champaign, Illinois, United States [3]Reflection AI [4]National Taiwan University, Taipei, Taiwan. Correspondence to: Ping-Chun Hsieh <pinghsieh@nycu.edu.tw>.

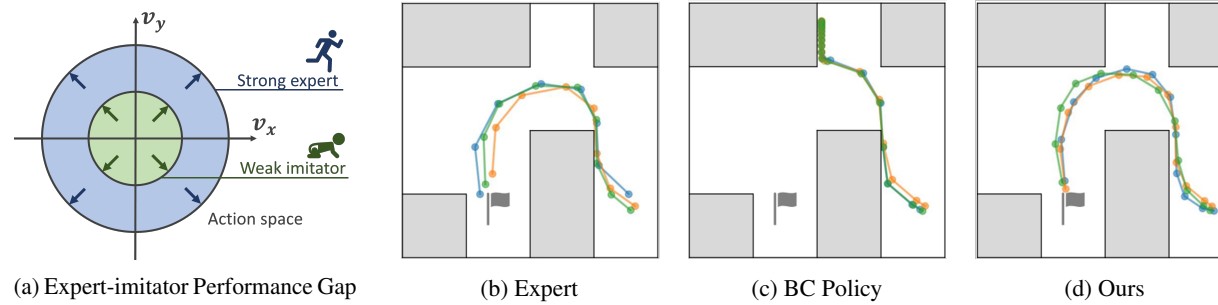

(a) Expert-imitator Performance Gap      (b) Expert      (c) BC Policy      (d) Ours

*Figure 1.* (a) illustrates the performance gap due to the mismatch between the action spaces of the strong expert and the weak imitator. (b) shows the expert trajectory in a Maze2d goal-reaching task, which serves as the reference for imitation. (c) demonstrates the trajectory of the action-constrained BC policy, which fails to execute a timely left turn due to the restricted action space. (d) depicts the trajectory generated by DTWIL alignment, where the agent adapts to the constrained action space by adjusting its pace, enabling it to accurately follow the expert trajectory.

sets. This issue leads to a problem we term "*occupancy measure distortion*." Figure 1 illustrates the issue of occupancy measure distortion caused by the mismatch between the action spaces of a strong expert and a weak imitator. The expert's trajectory highlights the optimal behavior, as shown in the data. However, the action-constrained behavior cloning policy fails to reproduce this trajectory due to the restricted action space, leading to suboptimal performance and collisions.

The most effective way to eliminate occupancy measure distortion is to ensure that both the expert and the imitator share the same feasible action set. However, since we cannot access the expert's policy to generate action-constrained expert demonstrations, we can only align to the expert's trajectories with constrained actions and learn from the alignment. This alignment is crucial to enable the imitator to accurately maintain consistency with the demonstrated trajectories. To address this challenge, we introduce Dynamic Time Warping Imitation Learning (DTWIL), an algorithm specifically designed to bridge this gap by generating surrogate action-constrained demonstrations and learning a corresponding policy that respects the constraints of the imitator. By aligning constrained trajectories with expert demonstrations, DTWIL provides a robust solution to mitigate occupancy measure distortion.

Our contributions can be summarized as follows: (1) We formalize the Action-Constrained Imitation Learning (ACIL) problem and identify the challenge of occupancy measure distortion. (2) We propose DTWIL, a novel framework for generating surrogate expert demonstrations and learning action-constrained policies. (3) We reformulate trajectory alignment as a planning problem, leveraging MPC for constrained trajectory generation. Moreover, we employ the DTW distance as the similarity metric and design practical implementations to enhance alignment efficiency. (4) Our experiments show that DTWIL outperforms the baseline IL algorithms in MuJoCo locomotion, navigation, and robot

arm manipulation tasks, excelling in sample efficiency and robustness to occupancy measure distortion.

## 2. Related Work

**Action-Constrained Reinforcement Learning (ACRL).** ACRL studies learn a policy under action constraints. Kasaura et al. (2023) provides benchmarks for evaluating existing ACRL approaches. Pham et al. (2018); Bhatia et al. (2019); Dalal et al. (2018) ensure safe and compliant behavior by incorporating a differentiable projection layer at the end of the policy network to meet action constraints. However, Lin et al. (2021); Brahmanage et al. (2023) highlight issues with this approach, particularly the zero gradient and longer training times, and propose alternative methods. Notably, Brahmanage et al. (2023); Chen et al. (2024) employ normalizing flows to directly generate actions that comply with the constraints, thereby circumventing the drawbacks associated with projection layers. Hung et al. (2025) proposes an efficient approach that significantly reduces the need for quadratic programs by using acceptance-rejection sampling and constructing an augmented Markov decision process. Yet, adapting existing ACRL techniques to ACIL is non-trivial.

**Learning from Demonstration.** IL focuses on deriving a policy using only the information from expert demonstrations, and this paradigm is also termed Learning from Demonstration (LfD). BC (Pomerleau & A, 1991) approaches this by treating a policy as a state-to-action mapping, learning it in a supervised manner. Adversarial Imitation Learning (AIL), on the other hand, focuses on matching the state-action distribution between expert and imitator through adversarial training. GAIL (Ho & Ermon, 2016) is a foundational method in this domain, using a discriminator to distinguish between expert and imitator transitions, and providing rewards based on this discrimination. Various AIL extensions (Kostrikov et al., 2019a;b; Lai et al.,

2024) improve upon GAIL, tailoring the method to different environments and goals. A comprehensive review of IL techniques can be found in (Zare et al., 2024), but ACIL, where a capability gap exists between the expert and the learner, remains unexplored.

**Learning from Observation (LfO).** An alternative approach to avoid the undesirable effects of the projected policy results after imitating expert actions is to learn only from expert observation data, which falls under the scenario of LfO (Lee et al., 2021; Huang et al., 2024). GAIfO (Torabi et al., 2018b) and IDDM (Yang et al., 2019) follow the principles of GAIL by training a state-only discriminator. OPOLO (Zhu et al., 2020) further improves on this by relaxing the on-policy requirement, speeding up the learning process. BCO (Torabi et al., 2018a) directly learns an inverse dynamics model to infer missing expert actions from observations and then applies BC to learn a policy. CFIL (Freund et al., 2023) and DIFO (Huang et al., 2024) use a generative model to capture state or state-action distributions. There also exists a line of work that focuses on state trajectory alignment. Liu et al. (2019); Gangwani & Peng (2020) aim to address the transition dynamics mismatch between the expert and the learner. Radosavovic et al. (2021) matches the state distributions in the context of dexterous manipulation, and Boborzi et al. (2023) proposes a non-adversarial framework for state-only imitation.

However, none of the existing LfO methods addresses the capability gap between the expert and the imitator. They attempt to mimic the trajectories induced by unconstrained actions, although such trajectories can be fundamentally infeasible for an action-constrained imitator to reproduce.

# 3. Action-Constrained Imitation Learning: Problem Statement and Challenges

In this section, we formalize the proposed ACIL problem and delineate the main challenges in achieving ACIL.

**Notation.** For any set $\mathcal{Z}$, we use $\Delta(\mathcal{Z})$ to denote the set of all probability distributions over $\mathcal{Z}$. Given any subset $\mathcal{X} \subset \mathbb{R}^n$ and any vector $x \in \mathbb{R}^n$, we let $\Gamma_{\mathcal{X}}(x)$ denote the $L_2$ projection of $x$ onto the set $\mathcal{X}$. For any sequence $x = (x_0, x_1, \cdots)$, we use $x_{t:t'}$ to denote the partial sequence $(x_t, \cdots, x_{t'})$.

## 3.1. Problem Statement

**Action-constrained Markov decision process.** We model the environment as an action-constrained Markov decision process (AC-MDP) defined by a tuple $\mathcal{M} := (\mathcal{S}, \mathcal{A}, \mathcal{T}, \mu, \gamma, \mathcal{C})$, where $\mathcal{S}$ and $\mathcal{A}$ are the state space and action space, respectively; $\mathcal{T} : \mathcal{S} \times \mathcal{A} \to \Delta(\mathcal{S})$ denotes the transition dynamics of the environment, with $\mathcal{T}(s'|s, a)$ in-

dicating the conditional distribution of the next state $s' \in \mathcal{S}$ given the current state $s \in \mathcal{S}$ and action $a \in \mathcal{A}$; $\mu \in \Delta(\mathcal{S})$ is the initial state distribution; $\gamma \in [0, 1)$ is the discount factor. For each state $s \in \mathcal{S}$, $\mathcal{C}(s) \subseteq \mathcal{A}$ denotes the feasible action set induced by the action constraints. Equivalently, under an AC-MDP, no actions that fall outside the feasible set $\mathcal{C}(s)$ can be employed to the environment at both training time and test time. Moreover, we impose no assumption on the structure of $\mathcal{C}(s)$, i.e., $\mathcal{C}(s)$ can be an arbitrary subset of the action space. Notably, the above setting of AC-MDP is standard in the ACRL literature (Lin et al., 2021; Kasaura et al., 2023; Brahmanage et al., 2023).

**Action-constrained imitation learning.** In ACIL, an action-constrained imitator is trained to perform similarly to an unconstrained expert from a set of expert demonstration trajectories $\mathcal{D}_e$ generated under an expert policy $\pi_e : \mathcal{S} \to \Delta(\mathcal{A})$. For ease of exposition, we let $\mathbb{T}$ and $\mathbb{T}_{\mathcal{C}}$ denote the sets of state-action trajectories that consist of unconstrained actions and constrained actions, respectively. Due to the gap in the action capability between the expert and the imitator, it is generally infeasible for the imitator to exactly reproduce an expert state-action trajectory $\tau_e \in \mathbb{T}$ or match the state-action distribution (also known as occupancy measure) of the expert. As a result, the standard IL methods, which focus on matching the state-action distributions, are not directly applicable in the action-constrained setting.

Therefore, in ACIL, our focus is on a more relaxed imitation learning setting: An action-constrained imitator learns to produce a state sequence as similar to the expert state sequence as possible. That is, in ACIL, the behavioral similarity is captured through *state sequence similarity*. Specifically, we use $\sigma(\tau) = (s_0, s_1, \cdots)$ to denote the state sequence induced by a state-action trajectory $\tau \in \mathbb{T}$ and let $\sigma_{t:t'}(\tau) := (s_t, \cdots, s_{t'})$ denote the partial sequence of $\sigma(\tau)$. Similarly, we use $a(\tau)$ to denote the action sequence included in $\tau$. Moreover, we let $\mathbb{S}$ denote the set of all possible state sequences (either of finite or countably infinite steps). For any pair of state sequences $\sigma, \sigma' \in \mathbb{S}$ (possibly of different lengths), we use $d(\sigma, \sigma')$ to denote the discrepancy between $\sigma$ and $\sigma'$, and the $d(\cdot, \cdot)$ is to be configured in the sequel. Our goal is to learn a feasible imitator policy $\pi^* \in \Pi_{\mathcal{C}}$ that minimizes the expected discrepancy from the expert, i.e.,

$$\pi^* := \arg\min_{\pi \in \Pi_{\mathcal{C}}} \mathbb{E}_{\tau_e \sim \pi_e, \tau \sim \pi} \big[ d(\sigma(\tau), \sigma(\tau_e)) | s_0 \sim \mu \big]. \quad (1)$$

## 3.2. Challenges of ACIL

Due to the action constraints, solving ACIL naturally involves the following two salient technical challenges compared to the conventional unconstrained IL settings:

**Aligning state sequences of different lengths.** To mimic

an expert state sequence of $T$ time steps, the imitator could possibly require more than $T$ steps to generate a similar state sequence, mainly due to the limited action capability. For example, as illustrated by the Maze2D navigation task in Figure 1, the imitator in Figure 1d takes several more steps than the expert to reach the goal state, despite their similarity in the state sequences. This issue would be more significant as the action constraints become tighter.

**Trajectory misalignment under projection.** To enforce the action constraints required by the imitator, one direct approach is to employ an off-the-shelf unconstrained IL algorithm and apply an additional action projection step, which identifies the nearest feasible action to the original unconstrained action generated by the policy. Notably, the action projection step has been widely adopted by the ACRL algorithms, either as a post-processing subroutine (Lin et al., 2021; Kasaura et al., 2023) or as a differentiable layer at the policy output for end-to-end training (Pham et al., 2018; Bhatia et al., 2019). Despite its effectiveness in ACRL, action projection can lead to severe trajectory misalignment as the action discrepancies could quickly compound the state discrepancies. Such trajectory misalignment has also been illustrated in Figure 1c. The misalignment issue further highlights that *ACIL and ACRL are fundamentally different problems in spite of their high-level resemblance.* As a result, ACIL cannot be tackled simply by action projection, and an alternative solution is needed.

# 4. Methodology

In this section, we formally introduce the proposed trajectory alignment method for addressing ACIL. We start by presenting the overall framework and then describe the two main modules, namely MPC-based trajectory optimization and dynamic time warping.

## 4.1. Surrogate Expert Demonstrations

To imitate the expert state sequence and enforce the action constraints simultaneously, we propose to decompose the imitation process of ACIL into two stages:

- **Stage 1: Generate action-constrained surrogate demonstrations from unconstrained expert demonstrations.** Given the original expert demonstrations $\mathcal{D}_e$, the main purpose here is to construct a surrogate dataset $\mathcal{D}_{sur}$ such that similar state sequences can be reproduced solely by using feasible actions. Specifically, for each state-action trajectory $\tau = (s_0, a_0, s_1, a_1, \cdots)$ in $\mathcal{D}_e$, we create a surrogate trajectory $\tau_{sur} = (s'_0, a'_0, s'_1, a'_1, \cdots)$ that satisfies: (i) $s'_0 = s_0$; (ii) $a'_i \in \mathcal{C}(s'_i)$, for all $i$; (iii) The discrepancy of the state sequences induced by $\tau$ and $\tau_{sur}$ is small.

- **Stage 2: Employ any IL method with surrogate demonstrations.** Given the surrogate dataset $\mathcal{D}_{sur}$ that involves only feasible actions, one can simply leverage any off-the-shelf IL algorithm to imitate the surrogate demonstrations to enforce the action constraints. In the subsequent experiments, we showcase this flexibility by using both BC and inverse RL methods to achieve effective ACIL.

Through this framework, we nicely decouple the constraint satisfaction from the downstream imitation learning method.

## 4.2. Trajectory Alignment via Model Predictive Control

**Trajectory alignment as trajectory optimization.** To construct the surrogate demonstrations (*i.e.*, Stage 1 described in Section 4.1), we propose to recast *trajectory alignment as a planning problem* and solve it by trajectory optimization. Specifically, for each expert state-action trajectory $\tau_e = (s_0, a_0, s_1, a_1, \cdots)$ in $\mathcal{D}_e$, the problem of finding a surrogate trajectory of feasible actions can be formulated as

$$\tau_{sur} = \underset{\tau \in \mathbb{T}_\mathcal{C}}{\arg\min} \ d(\sigma(\tau), \sigma(\tau_e)), \qquad (2)$$

where $d(\cdot, \cdot)$ is the state sequence discrepancy defined in Section 3.1. If we focus on the sequence of actions in $\tau_{sur}$ (denoted by $a_0^*, a_1^*, \cdots$), the problem in (2) can be reformulated as a trajectory optimization problem

$$a_{0:K^*-1}^* := \underset{\tau : a(\tau) \in \mathcal{C}^K, K \geq 1}{\arg\min} \ d(\sigma(\tau), \sigma(\tau_e)), \qquad (3)$$

where the minimization is over all feasible action sequences. Notably, directly solving (3) can be difficult for two reasons: (i) The optimal length of the surrogate trajectory (*i.e.*, $K^*$ in (3)) can be hard to determine a priori. (ii) Even if $K^*$ is known, it is known that directly handling a large planning problem of large length $K^*$ in a non-adaptive manner can be rather sub-optimal.

**Model predictive control for alignment.** To adaptively solve (3), we leverage MPC, which addresses (3) by sequentially solving finite-horizon planning subproblems with the help of a forward dynamics model. Specifically, we use MPC to determine the surrogate action at each step sequentially. For each step $t$ of $\tau_{sur}$, MPC configures a fixed planning horizon $H$ and approximately minimizes a finite-horizon objective function $d(\sigma(\tau'), \sigma_{t:t+H}(\tau_e))$ by first (i) generating a set of $H$-step synthetic rollouts $\{\tau'\}$ with the help of a forward dynamics model, (ii) selecting the rollout with the smallest $d(\sigma(\tau'), \sigma_{t:t+H}(\tau_e))$, and then (iii) assigning $a_t^*$ to be the first action of the selected rollout. This $a_t^*$ would be applied to the environment to obtain the next state. This design of taking only the first action ensures that each step of the alignment process can better adapt to the remaining part of the demonstration.

## 4.3. Dynamic Time Warping as the Alignment Criterion

To substantiate the ACIL objective function in Equation (1) and the trajectory optimization problem in Equation (3), we leverage the DTW distance (Hiroaki & Chiba, 1978) as the state sequence discrepancy metric. Recall from Section 3.2 that one main challenge of ACIL is the need to align state sequences of different lengths, and DTW naturally addresses this issue as a distance that can "warp" time over the space of time sequences.

We describe the concept of DTW in the context of ACIL as follows: Consider any two state sequences $\sigma, \sigma' \in \mathbb{S}$ of lengths $m$ and $n$. Let $\boldsymbol{\Delta}$ be the pairwise $\ell_2$ distance matrix of size $m \times n$, where $\boldsymbol{\Delta}_{i,j} = \|\sigma_i - \sigma'_j\|_2$. We use $\mathbf{A}$ to denote an $m \times n$ alignment matrix, which satisfies the following three properties: (i) $\mathbf{A}$ is a binary matrix, *i.e.*, it only has elements 0 or 1; (ii) The ones in $\mathbf{A}$ shall induce a path from the top-left to the bottom-right of the matrix; (iii) This path only involves three possible moves, *i.e.*, right, down, and lower right. We also let $\mathbb{A}$ be the set of all such matrices. Then, the DTW distance between $\sigma$ and $\sigma'$ is defined as

$$d_{\text{DTW}}(\sigma, \sigma') := \min_{\mathbf{A} \in \mathbb{A}} \langle \boldsymbol{\Delta}, \mathbf{A} \rangle_{\text{F}}, \qquad (4)$$

where $\langle \cdot, \cdot \rangle_{\text{F}}$ denotes the Frobenius inner product. Moreover, it is known that the DTW distance in Equation (4) can be found by dynamic programming through the following recursion

$$d_{\text{DTW}}(\sigma_{0:i}, \sigma'_{0:j}) = \|\sigma_i - \sigma_j\|_2 + \min \{ d_{\text{DTW}}(\sigma_{0:i-1}, \sigma'_{0:j}),$$
$$d_{\text{DTW}}(\sigma_{0:i}, \sigma'_{0:j-1}), d_{\text{DTW}}(\sigma_{0:i-1}, \sigma'_{0:j-1}) \}, \qquad (5)$$

where the minimum operation reflects the three possible moves (*i.e.*, right, down, and lower right) described above.

## 4.4. Putting Everything Together: DTWIL

We are ready to present the the proposed algorithm, namely Dynamic Time Warping Imitation Learning. We substantiate the generation process of surrogate demonstrations described in Section 4.1 by integrating MPC-based trajectory alignment with the DTW distance. Specifically: (i) *Alignment target*: We choose one expert trajectory $\tau^i \in \mathcal{D}_e$ at a time as the alignment target. (ii) *MPC-based alignment with DTW*: At the beginning of an alignment episode, MPC is initialized to the 0-th state of the selected expert trajectory. After initialization, MPC aligns with the target trajectory at each step sequentially until the episode is finished. To utilize DTW as the criterion, we introduce a *progression parameter*, $t_{\text{pg}}$, which indicates the timestep of the expert state with which the action-constrained agent is currently aligned. For example, if the current progress is at $t_{\text{pg}}$ and the planning horizon is set to $H$, the targeted

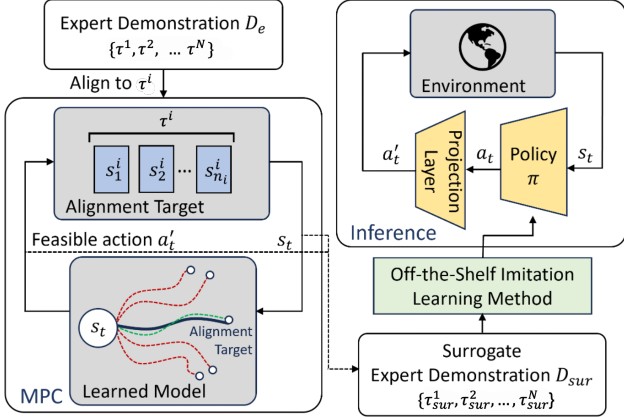

*Figure 2.* An illustration of the DTWIL framework. MPC aligns the action-constrained trajectories with expert demonstrations, generating surrogate demonstrations for downstream imitation (e.g., behavior cloning). The trained policy is then deployed during inference with actions projected onto the feasible action space.

segment of the expert state sequence for alignment would be $\sigma_{t_{\text{pg}}:(t_{\text{pg}}+H)}(\tau^i)$. Suppose the current MPC timestep is $t$, the current progress is $t_{\text{pg}}$, and the planning horizon is $H$. Then, MPC approximately finds an $H$-step planning trajectory $\tau'$ such that $d_{\text{DTW}}(\sigma(\tau'), \sigma_{t_{\text{pg}}:(t_{\text{pg}}+H)}(\tau^i))$ is minimized by generating a set of synthetic rollouts with the help of the learned dynamics model.

These surrogate expert demonstrations are then used to train a policy via any off-the-shelf IL method, *e.g.*, BC. An overview of DTWIL is provided in Figure 2, and the pseudo code can be found in Algorithm 1 and Algorithm 2.

*Remark* 4.1. In typical RL planning tasks, the objective function of MPC is to maximize the cumulative reward (Chua et al., 2018; Hansen et al., 2022). By contrast, in the ACIL setting, we use DTW distance as the criterion in aligning the state sequences of the expert and the surrogate one.

*Remark* 4.2. To facilitate the rollout generation in MPC, we also incorporate the cross-entropy method (CEM) optimizer, which iteratively refines the search for optimal actions by sampling, evaluating, and updating the distribution of candidate actions (Chua et al., 2018). Moreover, to ensure that the candidate synthetic rollouts in MPC are all feasible, we employ rejection sampling to enforce the action constraints. Other details about our MPC implementation can be found in Appendix A.1.

## 4.5. Practical Implementation

### 4.5.1. PROGRESSION MANAGEMENT

The progression parameter, $t_{\text{pg}}$, is initialized to 0 at the start of every trajectory alignment, indicating that the alignment begins from the 0-th state of the expert trajectory. At the beginning of each alignment step, we update $t_{\text{pg}}$ by analyzing

the warping map to determine how many expert states the agent's action has advanced. Consequently, if the agent's first planning state, $s_1$, is not sufficiently close to the next expert state, $s_1^e$, it is more likely to be matched with the current expert state, $s_0^e$. Figure 3 shows how this advancement value is determined. The value of $t_{pg}$ is updated after every MPC step. For how this mechanism affects the alignment, please refer to Appendix A.6.

---

**Algorithm 1** Dynamic Time Warping Imitation Learning

---

1: **Input:** Expert demonstration $\mathcal{D}_e = \{\tau^i\}_{i=1}^N$, number of episodes $M$
2: Surrogate demonstration dataset $\mathcal{D}_{sur} \leftarrow \{\}$
3: Dynamics model training dataset $\mathcal{D} \leftarrow \mathcal{D}_e$
4: **for** iteration $m = 1$ to $M$ **do**
5:      Select an expert trajectory $\tau^i$ from $\mathcal{D}_e$
6:      Train the dynamics model ensemble with $\mathcal{D}$
7:      $\tau_{sur}^i \leftarrow$ Trajectory Alignment$(\tau^i)$
8:      $\mathcal{D} \leftarrow \mathcal{D} \cup \tau_{sur}^i$
9:      **if** $d_{DTW}(\tau_{sur}^i, \tau^i) < d_{DTW}(\mathcal{D}_{sur}[i], \tau^i)$ **then**
10:          $\mathcal{D}_{sur}[i] \leftarrow \tau_{sur}^i$
11:      **end if**
12: **end for**
13: Learn an imitator policy from $\mathcal{D}_{sur}$ by an off-the-shelf imitation learning algorithm

---

**Algorithm 2** Trajectory Alignment

---

1: **Input:** Planning horizon $H$, ERC horizon $h_{erc}$, $i$-th expert trajectory $\tau^i = \{(s_t^i, a_t^i)\}_{t=0}^{length=l} \in \mathcal{D}_e$, action constraint $\mathcal{C}$.
2: **Output:** $\tau_{sur}^i \leftarrow \{\}$
3: Agent's initial state $s_0 \leftarrow s_0^i$, $t_{pg} \leftarrow 0$
4: **for** time step $t = 0, 1, \ldots$ **do**
5:      **for** Actions sampled $a_{t:t+H}$ from CEM, 1 to NSamples **do**
6:          Apply $ERC(a_{t:t+H}, H, h_{erc}, \mathcal{C})$
7:          Propagate $\tau'$ from $s_t$ using dynamics models
8:          Evaluate $a_{t:t+H}$ as $d_{DTW}(\sigma(\tau'), \sigma_{t_{pg}:(t_{pg}+H)}(\tau^i))$
9:          Update CEM distribution
10:      **end for**
11:      Execute $a_t^*$
12:      $\tau_{sur}^i \leftarrow \tau_{sur}^i \cup (s_t, a_t^*)$
13:      **if** Progression has advanced in the warping path **then**
14:          $t_{pg} \leftarrow t_{pg} + 1$
15:      **end if**
16: **end for**

---

### 4.5.2. EXPERT REGULARIZED CONTROL

In environments that require precise movements, even small errors can cause significant disturbances. Inspired by Actor Regularized Control (ARC) (Sikchi et al., 2021), we mix

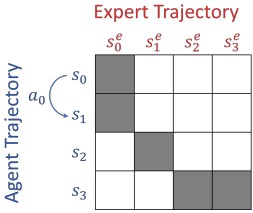
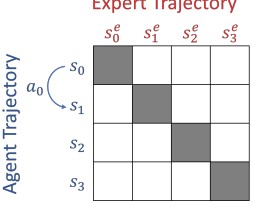

(a) Progression Advancement = 0      (b) Progression Advancement = 1

*Figure 3.* Since the MPC controller executes only the first planning step per iteration, we focus on the number of expert states the agent advances after the initial action $a_0$. The figure shows two DTW warping path cases (patches in gray). In Figure 3a, the agent transitions from $s_0$ to $s_1$ while staying aligned with $s_0^e$ causing no progression ($t_{pg}$ unchanged). In Figure 3b, the agent advances to the next expert state, updating $t_{pg}$ to $t_{pg} + 1$.

expert actions with sampled actions to serve as guidance, a mechanism that we refer to as Expert Regularized Control (ERC). Specifically, the first few actions used to roll out the planning trajectories in the MPC controller become the weighted average of the actions sampled and the corresponding segment of the actions of the experts. The details of the implementation can be found in Appendix A.7.

For other implementation details, please refer to Appendix A.4 and Appendix A.5.

## 5. Experiments

We evaluate our proposed method in various continuous control domains, including navigation, locomotion, and robot arm manipulation, subject to a variety of action constraints.

### 5.1. Setup

**Environments.** To evaluate our method, we conducted experiments on four benchmark tasks: *Maze2d*, *HalfCheetah*, *Hopper*, and *Table-Wiping*. Maze2d-Medium-v1 (Fu et al., 2020), a point-mass agent navigates a 2D maze from a random start location to a goal, with a 2-dimensional action space $[a_1, a_2] \in [-1.0, 1.0]$. The state information includes $v_1$ and $v_2$, representing the agent's velocity in the $x$- and $y$-directions. We collected 100 demonstrations, yielding 18,525 state-action pairs. In HalfCheetah (Brockman et al., 2016), a bipedal cheetah runs forward by applying torque through a 6-dimensional action space $[a_1, a_2, \ldots, a_6] \in [-1.0, 1.0]$. The state includes $w_i$, the angular velocity of each joint. For this task, we use 5 expert demonstrations of 1000 steps each. In Hopper (Brockman et al., 2016), a robot hops forward by controlling a 3-dimensional action space $[a_1, a_2, a_3] \in [-1.0, 1.0]$. Similarly, the state contains $w_i$, the angular velocity of each

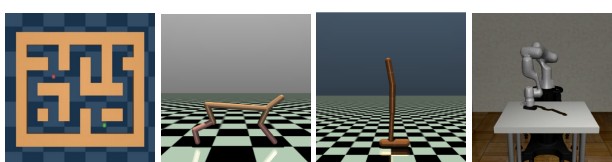

*Figure 4.* We evaluate the impact of action constraints on DTWIL and baseline methods across three environments: Maze2d-Medium-v1, HalfCheetah-v3, Hopper-v2, and Robosuite Table-Wiping task.

*Table 1.* Experiment environments and constraints

| Environment | Task | Constraint |
|---|---|---|
| Maze2d | M+B | $\lvert a_i \rvert < 0.1, \quad \forall i$ |
| | M+O | $\sum_{i=1}^{2} \lvert v_i a_i \rvert \leq 0.5$ |
| Hopper | H+B | $\lvert a_i \rvert < 0.9, \quad \forall i$ |
| | H+M | $\sum_{i=1}^{3} \max\{w_i a_i, 0\} \leq 10$ |
| HalfCheetah | HC+B | $\lvert a_i \rvert < 0.5, \quad \forall i$ |
| | HC+O | $\sum_{i=1}^{6} \lvert w_i a_i \rvert \leq 10$ |
| Table-Wiping | W+B | $\lvert a_i \rvert < 0.3, \quad \forall i$ |
| | W+L2 | $a_1^2 + a_2^2 + a_3^2 \leq 0.5,$ $a_4^2 + a_5^2 + a_6^2 \leq 0.05$ |

joint. For Hopper, we use 5 expert demonstrations of 1000 steps each for training. In Table-Wiping from Robosuite (Zhu et al., 2025), a robot arm controlled by a 6-dimensional action space aims to wipe a stained table. The expert dataset consists of ten 500-step demonstrations. The constraints considered for each environment are summarized in Table 1, including box constraints (+B), state-dependent power limit constraints (+M and +O), and L2 constraints (+L2).

**Baselines.** We compare our method to various baselines in learning from demonstration (LfD), *i.e.*, state-action sequences, and learning from observation (LfO), *i.e.*, state-only sequences.

- **BCO** (Torabi et al., 2018a) is an LfO method, learning an inverse dynamics model to infer actions from state-only data and applying BC to learn a policy.

- **GAIL** (Ho & Ermon, 2016) is an LfD method that utilizes a generative adversarial network (GAN) to infer the underlying reward function.

- **GAIfO** (Torabi et al., 2018b) resembles the idea of GAIL, but its discriminator only learns from state transitions $(s, s')$ instead of state-action pairs $(s, a)$.

- **OPOLO** (Zhu et al., 2020) is an off-policy LfO method and is among the most effective LfO techniques.

- **SAIL** (Liu et al., 2019) learns a reward function to align the agent and the expert state trajectories.

- **CFIL-s & CFIL-sa** (Freund et al., 2023) utilize a flow-based model to capture state or state-action distributions. CFIL-s learns from state-only sequences (LfO), while CFIL-sa learns from both states and actions (LFD).

- **DIFO** (Huang et al., 2024) learns a diffusion discriminator and learns the policy with the diffusion reward.

Note that we project output actions to feasible sets with a projection layer to avoid constraint violation for all methods.

### 5.2. Experimental Results

Since we aim to investigate sample efficiency, we only allow 50K environment steps during the training of all the online methods, including ours, on all tasks. All results are evaluated with randomly initialized starting states. Following this, the best-performing model from each algorithm during these interactions was selected for final evaluation. This ensures that the results reflect the effectiveness of each method in a small-sample regime.

The experimental results in Table 2 indicate that online algorithms, such as GAIL and OPOLO, face two primary challenges: poor sample efficiency and action constraints, leading to consistently suboptimal performance in all tasks. CFIL also struggles to achieve competitive performance under limited sample conditions. Despite their ability to interact with the environment, these methods fail to bridge the expert-imitator performance gap within the given number of interaction steps, resulting in persistently low scores. While BCO shows competitive performance in simpler tasks like Maze2d M+O, it falls short in more complex environments.

In contrast, DTWIL, which learns from surrogate expert data and adopts a BC approach to learn the policy, performs well in all tasks. By learning from the surrogate demonstrations to match the expert trajectories and using BC for policy learning, DTWIL manages to replicate expert performance while maintaining sample efficiency. As a result, it successfully reproduces expert-like trajectories across tasks, without being adversely affected by the constraints that cripple other methods. The results of training the various baseline methods for sufficient steps are included in Appendix A.3.

### 5.3. Ablation Study

**DTW distance and $\ell_2$ distance as alignment criteria.** We investigate the impact of using different alignment criteria, specifically comparing $\ell_2$ distance and DTW distance, on the performance of the DTWIL framework. The $\ell_2$ distance evaluates the pointwise differences between trajectories, while the DTW distance captures temporal misalignment, providing a more flexible measure for trajectory alignment.

*Table 2.* Evaluation performance of the proposed method and baseline algorithms across various tasks, with results expressed as the mean and standard deviation calculated from three seeds.

| Task | Metric | GAIL | BCO | GAIfO | OPOLO | CFIL-s | CFIL-sa | SAIL | DIFO | DTWIL (Ours) |
|---|---|---|---|---|---|---|---|---|---|---|
| M+B | return | 0.22 ± 0.0 | 0.14 ± 0.05 | 0.07 ± 0.02 | 0.2 ± 0.06 | 0.23 ± 0.06 | 0.23 ± 0.21 | 0.45 ± 0.06 | 0.27 ± 0.04 | **0.77 ± 0.04** |
| | $d_{\mathrm{DTW}}$ | 10.8 ± 0.4 | 7.8 ± 0.5 | 7.3 ± 0.2 | 8.2 ± 0.5 | 10.5 ± 0.6 | 10.9 ± 1.8 | 16.0 ± 0.61 | 7.2 ± 0.1 | **4.0 ± 0.1** |
| M+O | return | 0.14 ± 0.05 | **0.88 ± 0.06** | 0.19 ± 0.08 | 0.64 ± 0.13 | 0.45 ± 0.12 | 0.47 ± 0.10 | 0.46 ± 0.15 | 0.45 ± 0.09 | **0.87 ± 0.04** |
| | $d_{\mathrm{DTW}}$ | 10.4 ± 0.8 | 8.8 ± 0.5 | 8.5 ± 0.2 | 8.8 ± 0.3 | 10.8 ± 1.1 | 10.2 ± 0.7 | 15.1 ± 0.3 | 7.2 ± 0.0 | **3.6 ± 0.1** |
| HC+B | return | -163 ± 47 | -4 ± 4 | -74 ± 32 | -605 ± 390 | -172 ± 738 | -95 ± 515 | 1222 ± 260 | 8 ± 60 | **2669 ± 4** |
| | $d_{\mathrm{DTW}}$ | 32.5 ± 0.0 | 58.7 ± 11.9 | 56.3 ± 6.6 | 39.5 ± 7.3 | 37.9 ± 3.0 | 56.2 ± 11.8 | 26.5 ± 0.4 | 43.7 ± 1.9 | **7.4 ± 0.0** |
| HC+O | return | -185 ± 66 | 6 ± 31 | -163 ± 33 | -9 ± 80 | 1422 ± 1830 | 1674 ± 1316 | **2666 ± 55** | -87 ± 55 | 2637 ± 26 |
| | $d_{\mathrm{DTW}}$ | 37.0 ± 0.7 | 43.1 ± 17.8 | 76.8 ± 6.5 | 35.1 ± 4.2 | 27.9 ± 1.9 | 43.0 ± 21.6 | 19.7 ± 0.1 | 44.4 ± 1.8 | **7.8 ± 0.4** |
| H+B | return | 360 ± 59 | 219 ± 20 | 197 ± 30 | 1068 ± 952 | 866 ± 249 | 1485 ± 677 | 1375 ± 790 | 331 ± 29 | **2844 ± 57** |
| | $d_{\mathrm{DTW}}$ | 36.3 ± 1.6 | 33.1 ± 2.1 | 33.2 ± 1.4 | 33.7 ± 2.8 | 32.9 ± 2.8 | 29.8 ± 4.2 | 21.7 ± 0.5 | 34.3 ± 1.9 | **5.8 ± 0.1** |
| H+M | return | 261 ± 81 | 224 ± 32 | 206 ± 19 | 228 ± 33 | 1443 ± 547 | 1553 ± 1096 | 1273 ± 495 | 333 ± 32 | **2873 ± 240** |
| | $d_{\mathrm{DTW}}$ | 37.1 ± 0.4 | 33.1 ± 2.2 | 32.4 ± 0.9 | 31.8 ± 0.9 | 29.4 ± 2.3 | 31.0 ± 4.2 | 24.9 ± 0.8 | 33.3 ± 1.2 | **6.6 ± 1.5** |
| W+B | return | 12 ± 6 | **59 ± 20** | 30 ± 14 | 29 ± 20 | 0 ± 0 | 1 ± 0 | **61 ± 9** | 6 ± 1 | **61 ± 3** |
| | $d_{\mathrm{DTW}}$ | 32.5 ± 2.6 | 29.6 ± 1.2 | 36.3 ± 2.7 | 38.8 ± 7.6 | 39.2 ± 8.9 | 38.4 ± 5.5 | 22.1 ± 3.9 | 37.5 ± 4.3 | **11.8 ± 2.3** |
| W+L | return | 19 ± 12 | **91 ± 17** | 21 ± 9 | 42 ± 34 | 0 ± 0 | 1 ± 1 | 58 ± 3 | 7 ± 1 | 70 ± 4 |
| | $d_{\mathrm{DTW}}$ | 37.9 ± 1.7 | 22.9 ± 1.3 | 30.1 ± 2.9 | 31.1 ± 4.3 | 34.6 ± 3.7 | 36.4 ± 6.2 | 19.9 ± 0.8 | 41.6 ± 2.4 | **9.6 ± 0.4** |

*Table 3.* Comparison of policy performance using $\ell_2$ distance and DTW distance as alignment criteria across different tasks.

| Task | $\ell_2$ | DTW |
|---|---|---|
| HC+B | 2157.52 ± 60.84 | **2669.41 ± 4.56** |
| HC+O | 2279.12 ± 51.18 | **2637.34 ± 26.82** |
| H+B | 1054.49 ± 227.64 | **2844.68 ± 57.77** |
| H+M | 1245.61 ± 47.05 | **2873.88 ± 240.46** |

*Table 4.* Performance comparison of BC and DTWIL.

| Task | BC | DTWIL (Ours) |
|---|---|---|
| M+B | 0.61 ± 0.05 | **0.77 ± 0.04** |
| M+O | 0.81 ± 0.05 | **0.87 ± 0.04** |
| H+B | 2204.83 ± 753.32 | **2844.68 ± 57.77** |
| H+M | 1233.96 ± 211.87 | **2873.88 ± 240.46** |

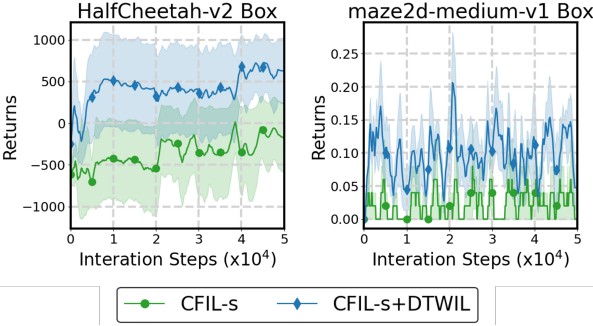

*Figure 5.* Performance comparison of CFIL with and without DTWIL-generated surrogate data across multiple tasks.

We conduct experiments on HalfCheetah and Hopper. For each task, we train policies using surrogate demonstrations generated with both alignment criteria and evaluate the policies on their ability to reproduce expert-like behavior.

The results, summarized in Table 3, demonstrate that the DTW distance significantly outperforms $\ell_2$ distance as an alignment criterion. Policies trained using DTW-aligned surrogate demonstrations achieve higher task performance and exhibit trajectories that more closely resemble expert behaviors. This highlights the importance of a flexible alignment metric in addressing temporal distortions and occupancy measure discrepancies in action-constrained environments.

**DTWIL-Augmented IL.** A key advantage of our approach is its versatility — the surrogate demonstrations generated by DTWIL can be integrated with any online or offline IL method. Table 4 compares our method to the original BC under action constraints, illustrating that DTWIL, which applies BC learning with surrogate data, consistently achieves superior performance across multiple tasks. Furthermore, we apply CFIL to the surrogate data and observe a notable performance boost, as shown in Figure 5. These results demonstrate that the surrogate data effectively mitigates occupancy measure distortion, a fundamental challenge in action-constrained imitation learning. They also suggest that our approach is broadly applicable and can enhance the performance of various IL methods in constrained environments.

## 6. Conclusion

ACIL has the potential to greatly influence real-world robot training, where constrained action spaces arise from power limits, mechanical imperfections, or wear and tear - chal-

lenges that previous methods have not effectively addressed. In this paper, we identify occupancy measure distortion as a key issue when learning from expert demonstrations under action constraints. We further introduce DTWIL, the first ACIL method, which leverages the DTW distance for alignment to generate surrogate expert demonstrations for downstream IL methods. DTWIL outperforms projection-based methods, demonstrating that a dedicated algorithm for the ACIL problem is both effective and necessary.

## Impact Statement

This paper presents work whose goal is to advance the field of Machine Learning. There are many potential societal consequences of our work, none of which we feel must be specifically highlighted here.

## Acknowledgements

This material is based upon work partially supported by the National Science and Technology Council (NSTC), Taiwan, under Contract No. NSTC 113-2628-E-A49-026, and the Higher Education Sprout Project of the National Yang Ming Chiao Tung University and the Ministry of Education, Taiwan. We also thank the National Center for High-performance Computing (NCHC) for providing computational and storage resources. We appreciate the financial support from the Featured Area Research Center Program within the framework of the Higher Education Sprout Project by the Ministry of Education (NTU-114L900901). Shao-Hua Sun was supported by the Yushan Fellow Program by the Ministry of Education, Taiwan.

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

# A. Detailed Implementation of DTWIL

## A.1. CEM Optimizer

Our implementation of the CEM optimizer closely follows the approach used in PETS (Chua et al., 2018), where a momentum term is added into the update calculations, and bounds are imposed on the standard deviations in addition to the standard CEM optimization.

Specifically, if a distribution at CEM iteration $i$, $\mathcal{N}(\mu_i, \sigma_i^2)$, is updated toward a target distribution $\mathcal{N}(\mu_{\text{target}}, \sigma_{\text{target}}^2)$, the resulting updated distribution at iteration $i + 1$, $\mathcal{N}(\mu_{i+1}, \sigma_{i+1}^2)$, will be given by:

$$\mathcal{N}(\mu_{i+1}, \sigma_{i+1}^2) = \mathcal{N}(\ \alpha\mu_i + (1-\alpha)\mu_{\text{target}},\ \alpha\sigma_i^2 + (1-\alpha)\sigma_{\text{target}}^2\ ),\ \alpha \in [0, 1]\,, \tag{6}$$

and the value of $\sigma_i^2$ is further constrained by $\frac{1}{2}w$, where $w$ represents the minimum distance from $\mu_i$ to the bounds of the feasible action space.

Moreover, to adapt the CEM optimizer for our action-constrained setting, we employ rejection sampling to ensure that all sampled actions strictly adhere to the predefined constraints.

## A.2. Dynamics Model

In this work, we train an ensemble of probabilistic neural networks to model the system's dynamics. Specifically, we utilize ensembles of five dynamics models, where the $b^{th}$ model, $f_{\theta_b}$, is parameterized by $\theta_b$. Each network in the ensemble is trained to minimize the negative log-likelihood of the predicted outcomes, optimizing the following objective:

$$\mathcal{L}(\theta_b) = -\sum_{n=1}^{N} \log f_{\theta_b}(s_{n+1}|s_n, a_n). \tag{7}$$

Referring to the ensembles used in PETS (Chua et al., 2018), we define our network to output a Gaussian distribution with diagonal covariance parameterized by $\theta$ and conditioned on $s_n$ and $a_n$, i.e.: $f = Pr(s_{t+1}|s_t, a_t) = \mathcal{N}(\mu_\theta(s_t, a_t), \sum_\theta(s_t, a_t))$. In this specific case, Equation (7) becomes:

$$\mathcal{L}_G(\theta_b) = \sum_{n=1}^{N} \left[\mu_{\theta_b}(s_n, a_n) - s_{n+1}\right]^\top \mathbf{\Sigma}_{\theta_b}^{-1}(s_n, a_n) \left[\mu_{\theta_b}(s_n, a_n) - s_{n+1}\right] + \log \det \mathbf{\Sigma}_{\theta_b}(s_n, a_n), \tag{8}$$

The next states are obtained in the same manner as $\mathbf{TS}\infty$ described in PETS.

Additionally, to mitigate the risk of over-fitting that can occur when a dynamics model is trained solely on expert trajectories, we augment the training data with online agent experiences and iteratively retrain the dynamics models.

## A.3. Training Curves for Baseline Methods with Additional Steps

In Section 5.2, we presented the performance of DTWIL and various baseline methods when interacting with the environment for up to 50K steps, focusing on sample efficiency. In Figure 6, we showcase the training curves of baseline methods over 500K steps, which is 10 times the original limit. These results reveal that methods like CFIL and OPOLO can train effective policies on multiple tasks when granted sufficient interaction steps. However, compared to DTWIL, which requires only the training of an MPC dynamics model to generate surrogate expert demonstrations, these online LfO methods demand significantly more interaction steps, highlighting their inefficiency relative to DTWIL.

## A.4. DTW Input Normalization

To address variations in scale across different dimensions, we normalize both the planned trajectory and the corresponding expert trajectory segment before computing the DTW distance. Specifically, we apply min-max normalization, which linearly transforms each dimension so that its values are scaled to fall within a consistent range. This is achieved by subtracting the minimum value and dividing by the range (maximum value minus minimum value) of the expert trajectory for each dimension. We analyze the impact of this normalization. Table 5 shows an ablation study on HalfCheetah and Hopper with their respective box constraints. We observe a performance drop in both environments when this normalization step is omitted from DTWIL. This is because, without normalization, DTW becomes disproportionately influenced by

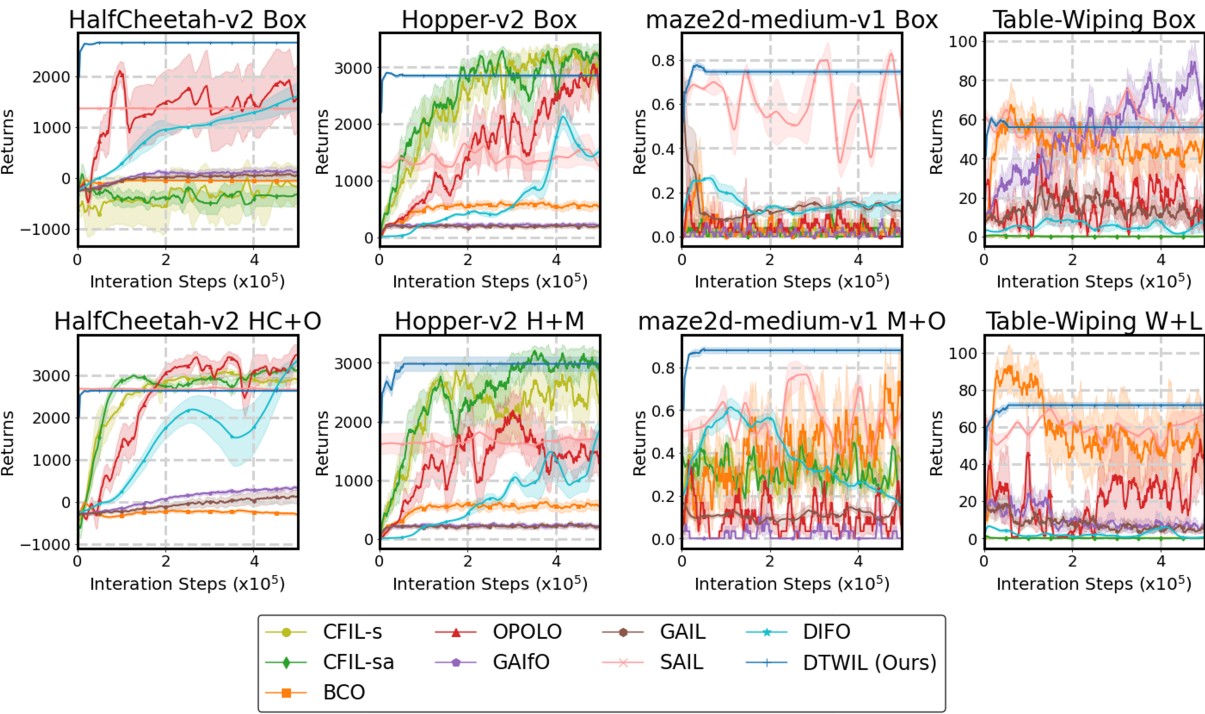

*Figure 6.* Training curves for baseline methods over 500K interaction steps across multiple tasks.

dimensions with larger scales, leading to poor generalization. Conversely, when the states are normalized in advance, DTW treats each dimension equally, resulting in more effective warping.

*Table 5.* Impact of DTW input normalization on performance. "W/o N" indicates results obtained without applying DTW input normalization.

| Task | HalfCheetah Box | HalfCheetah Box w/o N | Hopper Box | Hopper Box w/o N |
|------|-----------------|------------------------|------------|-------------------|
| Return-S | **2576.20 ± 61.62** | 1667.46 ± 51.13 | **2527.63 ± 572.53** | 608.18 ± 208.20 |
| Return-BC | **2669.41 ± 4.56** | 1893.90 ± 71.56 | **2844.68 ± 57.77** | 281.13 ± 31.88 |

### A.5. Excluding the Final Expert State

Notably, when constructing the warping path, the final expert state in the segment is excluded from the matching calculation to prevent unintended progression when the agent exhibits minimal movement across consecutive actions. Specifically, when two trajectories have an equal number of states, DTW often tends to align states in a strictly 1-to-1 manner, which can mislead progression. By excluding the final expert state, the DTW algorithm is encouraged to create a 2-to-1 alignment during the matching process. Given the constrained actions, which naturally take smaller steps than expert actions, this 2-to-1 alignment often occurs in the initial few states. This concept is illustrated in Figure 7. Including the final expert state (Figure 7a) leads to a 1-to-1 alignment since both trajectories have the same number of states. Excluding it Figure 7b prevents state from advancing, yielding a more desirable matching.

As demonstrated in Table 6, this adjustment significantly enhances performance in the Maze2d-Medium environment under box constraints. Specifically, excluding the final expert state when determining the DTW warping path improves the returns obtained during both the trajectory alignment phase and the subsequent behavioral cloning (BC) phase. These results validate the effectiveness of the proposed modification in stabilizing and optimizing the alignment process.

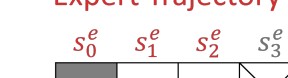

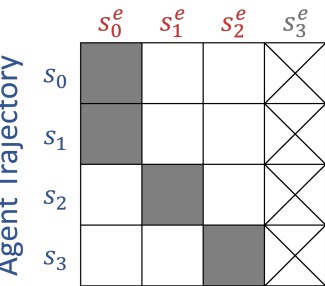

(a) Final expert state is not excluded    (b) Final expert state is excluded

*Figure 7.* Effect of excluding the final expert state on the DTW warping path.

*Table 6.* Results comparison of whether the final expert state is excluded when calculating the warping path in Maze2d-Medium under the box constraint.

|  | Excluded | Not Excluded |
|---|---|---|
| DTW-S | 2.99 ± 0.75 | 2.99 ± 0.82 |
| Return-S | **0.76 ± 0.00** | 0.69 ± 0.00 |
| Return-BC | **0.77 ± 0.04** | 0.72 ± 0.03 |

### A.6. Progression Management

In this section, we compare two approaches to progression management. The first is asynchronous progression, where the parameter $t_{pg}$ is updated in tandem with the warping path. This method is used in our algorithm. The second is synchronous progression, where $t_{pg}$ increases by 1 with each step of the imitator, matching the expert's pace. Given that agents with action constraint typically take longer to replicate expert behavior, asynchronous progression is more sensible. Table 7 presents the full experimental results for both methods. DTW-S denotes the DTW distance between the generated surrogate trajectories and the expert trajectories, Return-S indicates the average return of the surrogate expert data, and Return-BC represents the average return of BC policy trained on this surrogate expert data. While the differences on HalfCheetah are minimal, asynchronous progression significantly outperforms on Hopper.

*Table 7.* Comparison of results between asynchronous and synchronous progression methods.

| Task | Metric | Asynchronous | Synchronous |
|---|---|---|---|
| HC+B | DTW-S | 15.17 ± 0.24 | **15.06 ± 0.12** |
|  | Return-S | 2576.20 ± 61.62 | **2590.31 ± 24.07** |
|  | Return-BC | **2669.41 ± 4.56** | 2594.28 ± 29.80 |
| H+B | DTW-S | **11.70 ± 6.02** | 27.68 ± 0.26 |
|  | Return-S | **2527.63 ± 572.53** | 418.73 ± 89.35 |
|  | Return-BC | **2844.68 ± 57.77** | 153.52 ± 1.20 |

*Table 8.* Comparison of results with and without ERC applied during action sampling in Hopper.

|  | **Without ERC** | **With ERC** |
|---|---|---|
| Return-S | 820.7 ± 84.8 | **2527.6 ± 572.5** |
| Return-BC | 889.7 ± 5.4 | **2844.7 ± 57.8** |

### A.7. Implementation Details of ERC

In DTWIL, ERC acts as a stabilizer to guide sampled actions in environments requiring precise control. To implement this, we first extract a specific segment $a^{\mathrm{e}}_{t_{pg}:(t_{pg}+h_{\mathrm{erc}})}$, from the expert actions $a^{\mathrm{e}}$, where $h_{\mathrm{erc}}$ is the horizon over which expert actions are blended. Then, given the dynamics model ensembles $f_\theta(s, a)$, a specific weight $\beta \in [0, 1]$, and the projection function $\Gamma_{\mathcal{C}(s)}(a)$, which projects an action $a$ onto a specific constrained action space $\mathcal{C}(s)$, ERC guides the trajectory generation with the following functions:

$$\text{For } h = 0, 1, .., H :$$
$$a^{\mathrm{e}}_{\mathrm{proj}} = \Gamma_{\mathcal{C}(s_h)}(a^{\mathrm{e}}_{t_{pg}+h})$$
$$a_h = \begin{cases} \beta * a^{\mathrm{e}}_{\mathrm{proj}} + (1 - \beta) * a^{\mathrm{sampled}}_h , & \text{if } h <= h_{\mathrm{erc}} \\ a^{\mathrm{sampled}}_h , & \text{if } h > h_{\mathrm{erc}} \end{cases}$$
$$s_{h+1} = f_\theta(s_h, a_h) \tag{9}$$

where $a_h$ is the $h^{\mathrm{th}}$ action step in an $H$-step planning trajectory, $a^{\mathrm{sampled}}_h$ is the $h^{\mathrm{th}}$ action directly sampled from a CEM optimizer, and $s_h$ is the $h^{\mathrm{th}}$ state of the planning trajectory.

The performance of our algorithm in environments where agents are highly susceptible to deviations—such as Hopper, where falling results in early termination—is significantly enhanced by incorporating ERC. Table 8 demonstrate a clear performance difference: without ERC, the agent frequently falls, leading to significantly lower rewards and shorter trajectories. In contrast, incorporating ERC stabilizes the agent's behavior, allowing it to generate surrogate trajectories of appropriate length and maintain consistent performance throughout the task. This highlights the importance of ERC in enabling robust and reliable imitation under action-constrained settings. Refer to Appendix A.8 for detailed hyperparameter tuning.

### A.8. Hyperparameters in ERC

We explore the influence of the hyperparameter $\beta$, which regulates the balance between expert actions and MPC-sampled actions in the ERC method. Additionally, we examine the effect of the horizon length $h_{\text{erc}}$, which determines how many steps to blend MPC-sampled actions with expert actions. We conducted experiments on the Hopper with H+M constraints, varying $\beta$ from 0 to 0.2 and $h_{\text{erc}}$ from 0 to 20, while keeping all other hyperparameters fixed at their optimal values identified in prior tuning. As shown in Table 9, setting $\beta$ to 0.05 results in the highest performance. A lower $\beta$ leads to instability in the sampled actions, while higher values negatively impact the MPC optimization process. Regarding $h_{\text{erc}}$, a value of 5 provides the best results. Extending the horizon does not improve performance, as expert actions taken too far in the future become less informative due to the action constraints.

*Table 9.* Impact of varying $\beta$ and $h_{erc}$ values on performance in the Hopper task with H+M constraints. The table highlights the optimal balance between expert actions and MPC sampling, showing the best-performing configurations for stability and action guidance.

| | $\beta = 0$ | $\beta = 0.02$ | $\beta = 0.05$ | $\beta = 0.1$ | $\beta = 0.2$ |
|---|---|---|---|---|---|
| Return-S | 820.71 ± 84.78 | 1492.97 ± 144.35 | **2527.63 ± 572.53** | 1657.47 ± 286.44 | 670.72 ± 328.28 |
| Return-BC | 889.65 ± 5.39 | 1138.85 ± 56.35 | **2844.68 ± 57.77** | 2167.30 ± 360.73 | 723.95 ± 345.70 |

| | $h_{\text{erc}} = 0$ | $h_{\text{erc}} = 5$ | $h_{\text{erc}} = 10$ | $h_{\text{erc}} = 20$ |
|---|---|---|---|---|
| Return-S | 820.71 ± 84.78 | **2527.63 ± 572.53** | 2425.25 ± 370.40 | 2166.99 ± 351.04 |
| Return-BC | 889.65 ± 5.39 | **2844.68 ± 57.77** | 2686.85 ± 135.64 | 2616.09 ± 102.90 |

### A.9. Number of MPC Steps per Planning

Our MPC implementation executes only the first step of the planned trajectory after each action sampling. This design allows the alignment process to adapt more precisely to the remaining expert trajectory at every timestep. However, re-planning at every step can be computationally expensive. A common strategy to reduce the time complexity is to execute multiple steps per planning cycle, thereby amortizing the planning cost over several environment steps. This introduces a trade-off between computational efficiency and alignment fidelity. We conducted an ablation study by varying the number of execution steps taken before re-planning. We report the results in Table 10. The results indicate that as the number of steps increases, the alignment performance degrades. In particular, in the Hopper environment, poor alignment resulting from infrequent re-planning significantly increases the risk of falling.

*Table 10.* Ablation study on the number of steps executed per MPC planning cycle. Executing more steps before re-planning reduces alignment quality, leading to degraded performance.

| Steps \ Task | HC+B | HC+O | H+B | H+M |
|---|---|---|---|---|
| 1 step | 2669 ± 4 | 2637 ± 26 | 2844 ± 57 | 2873 ± 240 |
| 3 steps | 2427 ± 0 | 2324 ± 28 | 810 ± 21 | 753 ± 43 |
| 5 steps | 1270 ± 464 | 2123 ± 6 | 579 ± 2 | 579 ± 10 |

### A.10. Different Level of Constraints

We have shown that the imitator may suffer from occupancy measure distortion under action constraints. Beyond the type of constraint, the tightness of the constraint bounds also plays a critical role, as it directly determines the size of the imitator's feasible action space. If the constraints are loose, their impact on the learner's ability to replicate expert trajectories is minimal. In the extreme case where the constraints do not restrict any of the expert's actions, direct imitation learning

can still achieve strong performance. In our main experiments, we adopt moderately tight constraints to ensure that they meaningfully affect the learning process. In this section, we further investigate the performance of our method under both looser and tighter constraint levels. Corresponding results are shown in Table 11. The experimental results show that across nearly all cases, DTWIL consistently outperforms baseline methods, demonstrating robustness to varying levels of constraint tightness.

*Table 11.* Ablation study on the strength of action constraints. We vary the constraint bounds across Maze2D (M+B) and HalfCheetah (HC+O) environments. Results show that DTWIL maintains strong performance under both looser and tighter constraint settings, indicating robustness to constraint tightness.

| Task (Constraint Level) | CFIL-s | CFIL-sa | BCO | OPOLO | DTWIL (Ours) |
|---|---|---|---|---|---|
| M+B (0.3) | $0.20 \pm 0.14$ | $0.21 \pm 0.03$ | $0.44 \pm 0.34$ | $0.44 \pm 0.16$ | **$0.86 \pm 0.04$** |
| M+B (0.1) – paper | $0.22 \pm 0$ | $0.23 \pm 0.21$ | $0.14 \pm 0.05$ | $0.20 \pm 0.06$ | **$0.77 \pm 0.04$** |
| M+B (0.07) | $0.05 \pm 0.02$ | $0.06 \pm 0.01$ | $0.26 \pm 0.10$ | $0.34 \pm 0.13$ | **$0.68 \pm 0.01$** |
| HC+O (15) | $2065 \pm 714$ | **$2796 \pm 247$** | $-151 \pm 13$ | $47 \pm 673$ | **$2785 \pm 17$** |
| HC+O (10) – paper | $1422 \pm 1830$ | $1674 \pm 1316$ | $6 \pm 31$ | $-9 \pm 80$ | **$2637 \pm 26$** |
| HC+O (7) | $54 \pm 671$ | $354 \pm 1587$ | $-130 \pm 9$ | $99 \pm 941$ | **$1408 \pm 25$** |

## A.11. Number of Expert Demonstrations

To evaluate the generalization ability of our method, we conduct an ablation study by varying the number of expert demonstrations used for training. Specifically, we investigate whether DTWIL can still learn effective policies when provided with fewer expert trajectories, which limits coverage of the initial state distribution.

As shown in Table 12, DTWIL consistently achieves strong performance across different demonstration sizes, indicating that it can generalize well even with limited access to expert data. This suggests the surrogate demonstrations generated by our method capture essential structure for learning, beyond merely replicating observed trajectories.

*Table 12.* Ablation study on the number of expert demonstrations. We vary the number of expert trajectories provided for training to evaluate the generalization ability of DTWIL. Results show that DTWIL maintains strong performance even with limited demonstrations, indicating its robustness and sample efficiency.

| Number of Demos \ Task | HC+B | H+M |
|---|---|---|
| 5 Demos | $2669 \pm 4$ | $2873 \pm 240$ |
| 3 Demos | $2585 \pm 26$ | $2994 \pm 35$ |
| 1 Demo | $2618 \pm 28$ | $2685 \pm 457$ |

