# OpenReview forum: "Action-Constrained Imitation Learning"
_ICML.cc/2025/Conference — ICML 2025 poster_

### Official Review · Reviewer_7mJi · 2025-03-11

**Overall Recommendation:** 3

**Summary:**

This work proposes Action-Constrained Imitation Learning (ACIL), a new problem setting in which an imitator learns from a constraint-free expert while operating under action constraints. The key challenge is the infeasible expert actions issue, and the authors solve the mismatch issue by trajectory alignment. The proposed method is DTWIL, which generates surrogate expert demonstrations by aligning constrained trajectories with expert ones using MPC and DTW distance. Experiments on MuJoCo and Robosuite tasks demonstrate superior sample efficiency and performance over standard IL methods.

**Claims And Evidence:**

DTWIL is claimed to mitigate the occupancy measure mismatch by aligning trajectories. Experimental results across multiple tasks, such as Maze2d, Hopper, HalfCheetah, and Robosuite, show that DTWIL outperforms baseline methods in terms of task performance and sample efficiency.

The authors argue that DTW is a more effective metric for trajectory alignment compared to the traditional l2 distance. Ablation studies confirm that policies trained with DTW-aligned trajectories achieve better performance.

**Essential References Not Discussed:**

N/A

**Experimental Designs Or Analyses:**

1. The experiments are designed to assess the effectiveness of DTWIL across various tasks with differing levels of action constraints.

2. The impact of using DTW v.s. l2 distance for trajectory alignment is analyzed, demonstrating the advantage of DTW.

**Methods And Evaluation Criteria:**

The proposed approach involves generating surrogate expert trajectories that adhere to the imitator's action constraints. This is formulated as a planning problem solved via MPC, with DTW used to measure trajectory similarity. And the evaluation metric is common.

**Other Comments Or Suggestions:**

N/A

**Other Strengths And Weaknesses:**

Strengths:
1. The introduction of ACIL addresses practical challenges in scenarios where imitators have inherent action limitations. The problem setting is interesting and practical.

2. The application of DTW for trajectory alignment in imitation learning is a novel approach that effectively handles temporal discrepancies between expert and imitator trajectories.

Weaknesses:
1. The author should make ablation results about the number of expert demos, and also discuss the different levels of action restrictions for the imitator.

2. The experiment on robotics is table-wiping, which is too simple. The authors are encouraged to conduct some manipulation tasks, at least picking and placing tasks.

3. No generalization experiments are discussed, which is significant for this setting.

**Questions For Authors:**

Why does DTW outperform the l2 distance to a large degree, can you give more proofs in theorems? Is it the optimization issue or the convergence issue?

**Relation To Broader Scientific Literature:**

The paper addresses a gap in imitation learning research by focusing on scenarios where the imitator operates under action constraints not present for the expert.

The use of DTW for aligning trajectories introduces a novel application of this technique in the context of imitation learning, differing from traditional methods that may not account for temporal variations in trajectories.

**Theoretical Claims:**

No theorems.

---

> ### Author Rebuttal · Authors · 2025-04-01
>
> We sincerely appreciate the reviewer’s insightful comments and suggestions. We provide our point-by-point response as follows.
>
> > The author should make ablation results about the number of expert demos, and also discuss the different levels of action restrictions for the imitator.
>
> Based on the reviewer’s suggestions, we conducted ablation studies on both the number of expert demonstrations and different levels of action restrictions for the imitator, testing these across two distinct environment settings. The results are shown in the tables below. Our findings demonstrate that our method remains strong under varying constraint levels and can generalize well to datasets with fewer expert demonstrations.
>
> Table 1: **Ablation Study on Different Expert Demos**
> | Number of Demos \ Task Name | HC+B | H+M |
> | -------- | -------- | -------- |
> | 5 Demo     | 2669 ± 4 | 2873 ± 240 |
> | 3 Demos     | 2585 ± 26 | 2994 ± 35 |
> | 1 Demos     | 2618 ± 28 | 2685 ± 457 |
>
> Table 2: **Ablation Study on Different Constraint Levels**
> | Task (Constraint Level) \ Method | CFIL-s | CFIL-sa | BCO | OPOLO | DTWIL |
> | -------- | -------- | -------- | -------- | -------- | -------- |
> | M+B (0.3)  | 0.20 ± 0.14 | 0.21 ± 0.03 | 0.44 ± 0.34 | 0.44 ± 0.16 | **0.86 ± 0.04** |
> | M+B (0.1) - paper  | 0.22 ± 0 | 0.23 ± 0.21 | 0.14 ± 0.05 | 0.20 ± 0.06 | **0.77 ± 0.04** |
> | M+B (0.07)    | 0.05 ± 0.02 | 0.06 ± 0.01 | 0.26 ± 0.10 | 0.34 ± 0.13 | **0.68 ± 0.01** |
> |||||||
> | HC+O (15) | 2065 ± 714 | **2796 ± 247** | -151 ± 13 | 47 ± 673 | **2785 ± 17** |
> | HC+O (10) - paper  | 1422 ± 1830 | 1674 ± 1316 | 6 ± 31 | -9 ± 80 | **2637 ± 26** |
> | HC+O (7)  | 54 ± 671 | 354 ± 1587 | -130 ± 9 | 99 ± 941 | **1408 ± 25** |
>
>
> > The experiment on robotics is table-wiping, which is too simple. The authors are encouraged to conduct some manipulation tasks, at least picking and placing tasks.
>
> We appreciate the reviewer's suggestion. However, we have observed that existing and even state-of-the-art ACRL works have not tackled robotic arm tasks either [1-3].Pick-and-place tasks are indeed more complex due to higher-dimensional action spaces and additional constraints such as grasp stability and object interactions[4]. Since this work is the first step in addressing ACIL, we focus on simpler settings and leave the challenge of extending ACIL to more complex robotic manipulation tasks for future work.
>
> [1] Janaka Brahmanage, Jiajing Ling, and Akshat Kumar, “FlowPG: Action-constrained policy gradient with normalizing flows,” NeurIPS 2023.
>
> [2] Kazumi Kasaura, Shuwa Miura, Tadashi Kozuno, Ryo Yonetani, Kenta Hoshino, and Yohei Hosoe, “Benchmarking actor-critic deep reinforcement learning algorithms for robotics control with action constraints,” Robotics and Automation Letters, 2023.
>
> [3] Hung, Wei, Shao-Hua Sun, and Ping-Chun Hsieh. "Efficient Action-Constrained Reinforcement Learning via Acceptance-Rejection Method and Augmented MDPs." arXiv preprint arXiv:2503.12932 (2025).
>
> [4] Lobbezoo, Andrew, Yanjun Qian, and Hyock-Ju Kwon. "Reinforcement learning for pick and place operations in robotics: A survey." Robotics 10.3 (2021): 105.
>
> > No generalization experiments are discussed, which is significant for this setting.
>
> We are unsure about the specific meaning of "generalization experiments" mentioned in this comment. However, we believe this may refer to the ablation study on varying numbers of expert demonstrations. In our experiments, we have tested two different environment settings, and the results are presented in Table 2 above.
>
> > Why does DTW outperform the l2 distance to a large degree, can you give more proofs in theorems? Is it the optimization issue or the convergence issue?
>
> DTW distance can be viewed as a special case of the Wasserstein distance, as discussed in Sec 2.1 in [5]. Specifically, when applied to discrete measures, the Wasserstein and DTW distances share the same form of optimization problem. In contrast, L2 distance computes element-wise differences without considering the overall alignment. Consequently, in sequence matching tasks, optimizing with L2 distance can lead to issues since it does not align with the underlying objective [6-7].
>
> [5] Samuel Cohen, Giulia Luise, Alexander Terenin, Brandon Amos, and Marc Peter Deisenroth, “Aligning Time Series on Incomparable Spaces,” AISTATS 2021.
>
> [6] David Schultz and Brijnesh Jain, “Nonsmooth Analysis and Subgradient Methods for Averaging in Dynamic Time Warping Spaces,” Pattern Recognition, vol. 74, pp. 340-358, 2018.
>
> [7] Rohit J. Kate, “Using Dynamic Time Warping Distances as Features for Improved Time Series Classification,” Data Mining and Knowledge Discovery, vol. 30, no. 1, pp. 1-22, 2015.

---

### Official Review · Reviewer_G927 · 2025-03-13

**Overall Recommendation:** 3

**Summary:**

This paper focues on a problem setting called Action-Constrained Imitation Learning. This setting involves an imitator agent that learns from an expert demonstrator, but the imitator has a more restricted action space than the expert. This difference in action capabilities creates a mismatch in the occupancy measure, making standard imitation learning methods ineffective.

To address this, the authors propose a new method called Dynamic Time Warping Imitation Learning. DTWIL works in two stages:

1. Surrogate Demonstration Generation: It replaces the original expert demonstrations with a "surrogate" dataset. This new dataset contains trajectories that are similar to the expert's state trajectories but adhere to the imitator's action constraints. This alignment is achieved by using MPC and Dynamic Time Warping distance. The MPC treats trajectory alignment as a planning problem, generating feasible actions for the imitator, while DTW serves as the similarity metric between the surrogate and expert trajectories.

2. Imitation Learning: Once the surrogate dataset is created, any standard imitation learning algorithm (like BC or IRL) can be used to train the imitator policy.

Experiments on robot control tasks show that DTWIL outperforms other imitation learning methods in this constrained setting.

## update after rebuttal

The authors addressed my major by adding an additional baseline, so I raised my rating.

**Claims And Evidence:**

I think most of the claims in this submission are supported by evidence. However, there are some concerns regarding their experimental evaluation, which will be discussed in sections below.

**Essential References Not Discussed:**

As mentioned in "Experimental Designs or Analyses," there is a line of work focusing on imitating states in the demonstrations rather than actions. This concept is also a core idea of this submission. Therefore, I believe these works should be discussed and incorporated into the experimental comparisons.

**Experimental Designs Or Analyses:**

- This paper overlooks an important set of baselines—state-based imitation learning [1,2,3,4]. In fact, the problem setting of "Action-Constrained Imitation Learning" is not entirely new and has been extensively discussed in these works. However, this submission does not discuss these papers. I think at least one of these methods should be **included as a baseline**. Without such a comparison, it is not entirely convincing that the proposed DTWIL method offers clear advantages over these existing approaches.

- I did not find a detailed explanation of how the action constraints are implemented in the simulator. If the weak imitator outputs an action that violates the action constraints, is it discarded or somehow mapped back to a valid action? This implementation detail could be crucial to certain design choices. Please let me know if I have overlooked this information in the paper.


[1] Liu, Fangchen, et al. "State Alignment-based Imitation Learning." International Conference on Learning Representations.

[2] Gangwani, Tanmay, and Jian Peng. "State-only Imitation with Transition Dynamics Mismatch." International Conference on Learning Representations.

[3] Radosavovic, Ilija, et al. "State-only imitation learning for dexterous manipulation." 2021 IEEE/RSJ International Conference on Intelligent Robots and Systems (IROS). IEEE, 2021.

[4] Boborzi, Damian, et al. "Imitation learning by state-only distribution matching." Applied Intelligence 53.24 (2023): 30865-30886.

**Methods And Evaluation Criteria:**

Yes, they make sense to me.

**Other Comments Or Suggestions:**

N/A

**Other Strengths And Weaknesses:**

### Strengths

- The proposed method DTWIL separates the concerns of constraint satisfaction (addressed during surrogate data generation) and imitation learning (which can then leverage any standard IL algorithm), offering greater flexibility in algorithm design.
- The introduction of Dynamic Time Warping (DTW) as a distance metric makes sense to me. DTW is particularly well-suited for comparing trajectories of varying lengths, which is essential in the ACIL setting.
- The writing is generally clear and accessible. The figures and tables are neat.


### Weaknesses

- The proposed method employs MPC to generate action-constrained surrogate demonstrations. However, MPC relies on the ability to reset the environment to specific states, which is feasible in simulation but not transferable to real-world scenarios. In contrast, many state-based imitation learning methods mentioned earlier do not have this requirement.
- One implicit assumption of this paper is that the state contains information strongly correlated with task success (e.g., the velocity of a cheetah or the position of the target object). Under this assumption, minimizing the discrepancy between the state sequences induced by the expert and the imitator is reasonable. However, a more common scenario involves access to only a partial state, meaning the task is a POMDP. It is unclear whether the proposed method would still perform effectively in such cases. Discussions addressing this point would be valuable.
- In Table 1, the meanings of "H+B," "M+O," "W+L2," etc., are not explained. I assume "B" refers to "box constraints," correct?

**Questions For Authors:**

See above sections.

**Relation To Broader Scientific Literature:**

See "Essential References Not Discussed".

**Theoretical Claims:**

This submission does not include proofs.

---

> ### Author Rebuttal · Authors · 2025-04-01
>
> We are grateful for the reviewer’s constructive feedback.
>
> >This paper overlooks an important set of baselines—state-based imitation learning [1,2,3,4]. In fact, the problem setting of "Action-Constrained Imitation Learning" is not entirely new and has been extensively discussed in these works. However, this submission does not discuss these papers.
>
> Thank you for your suggestion. We will include a discussion on this line of work in the main text. [1] and [2] focus on handling transition dynamics mismatches in state-only IL. [3] also uses state distribution matching to align the agent’s state trajectories with the expert's, and it’s more specific for dexterous manipulation tasks. [4] proposes a non-adversarial LfO approach. These works align the state-trajectories induced by **unconstrained actions**—trajectories that the constrained imitator cannot perfectly replicate, thus failing to achieve optimal performance. We found SAIL [1] the most relevant to our problem formulation, so we chose SAIL as a baseline. To adapt SAIL to the ACIL setting, we projected the actions generated by the SAIL policy onto each task’s feasible set before interacting with the environment, like other baselines. The table shows the return of SAIL on benchmark tasks. SAIL is only comparable to DTWIL in the HC+O and W+B task, while in all other tasks, it fails to achieve good returns.
>
> |Task Name|M+B|M+O|HC+B|HC+O|H+B|H+M|W+B|W+L2|
> |-|-|-|-|-|-|-|-|-|
> |SAIL|0.45±0.06|0.46±0.15|1222±260|**2666±55**|1375±790|1273±495|**61±9**|58±3|
> |**DTWIL**|**0.77±0.04**|**0.87±0.04**|**2669±4**|**2637±26**|**2844±57**|**2873±240**|**61±3**|**70±4**|
>
> >I did not find a detailed explanation of how the action constraints are implemented in the simulator. If the weak imitator outputs an action that violates the action constraints, is it discarded or somehow mapped back to a valid action?
>
> We follow the standard approach in the ACRL literature to enforce action constraints [5-7]. Specifically, we use a projection layer that ensures all output actions are mapped to the feasible set before interacting with the environment. This is mentioned in L364–L366, C1, and the caption of Figure 2. We appreciate the reviewer’s feedback and will add further clarification in the main text.
>
> [5] Janaka Brahmanage, Jiajing Ling, and Akshat Kumar, “FlowPG: Action-constrained policy gradient with normalizing flows,” NeurIPS 2023.
>
> [6] Kasaura, Kazumi, et al. "Benchmarking actor-critic deep reinforcement learning algorithms for robotics control with action constraints." IEEE Robotics and Automation Letters 8.8 (2023): 4449-4456.
>
> [7] Lin, Jyun-Li, et al. "Escaping from zero gradient: Revisiting action-constrained reinforcement learning via Frank-Wolfe policy optimization." Uncertainty in Artificial Intelligence. PMLR, 2021.
>
> >The proposed method employs MPC to generate action-constrained surrogate demonstrations. However, MPC relies on the ability to reset the environment to specific states, which is feasible in simulation but not transferable to real-world scenarios.
>
> The reviewer raises a valid concern regarding MPC's reliance on environment resets when using true dynamics for planning. However, our method mitigates this limitation by leveraging a learned dynamics model for planning, as done by various existing MPC methods [8-9]. Instead of requiring resets, we roll out possible trajectories within this model, making our approach more adaptable to real-world scenarios. Further details can be found in PETS [10].
>
> [8] Draeger, Andreas, Sebastian Engell, and Horst Ranke. "Model predictive control using neural networks." IEEE Control Systems Magazine 15.5 (1995): 61-66.
>
> [9] Kamthe, Sanket, and Marc Deisenroth. "Data-efficient reinforcement learning with probabilistic model predictive control." International conference on artificial intelligence and statistics. PMLR, 2018.
>
> [10] Chua, Kurtland, et al. "Deep reinforcement learning in a handful of trials using probabilistic dynamics models." Advances in neural information processing systems 31 (2018).
>
> >One implicit assumption of this paper is that the state contains information strongly correlated with task success. However, a more common scenario involves access to only a partial state, meaning the task is a POMDP. It is unclear whether the proposed method would still perform effectively in such cases.
>
> Since even standard MDP settings are not extensively discussed in prior work, we take the first step towards tackling action-constrained imitation learning and leave the challenges of partial observability to future work. One possible direction to address this issue is to learn a latent representation that captures task-relevant information, enabling the agent to infer missing state information.
>
> >In Table 1, the meanings of "H+B," "M+O," "W+L2," etc., are not explained. I assume "B" refers to "box constraints," correct?
>
> The reviewer is correct here. "B" refers to "box constraints." We will add explanations to the main text.

---

> > ### Comment · Reviewer_G927 · 2025-04-05
> >
> > I appreciate the additional baseline; please include it in your manuscript. I have updated my rating accordingly.

---

> > > ### Author Response · Authors · 2025-04-05
> > >
> > > We thank the reviewer again for all the insightful comments and the time put into helping us to improve our submission. We will include the additional baseline in the final version.

---

### Official Review · Reviewer_Fdv6 · 2025-03-14

**Overall Recommendation:** 3

**Summary:**

This paper explores a new task, Action-Constrained Imitation Learning, that the imitator may have the constrained action space compared with export. Then, this paper analyze the challenge of this task and propose a methods to solve this challenge. Specially, this paper first employ MPC to build a surrogate dataset, where the trajectories in this dataset are similar with the export and with same action space with imitator.

**Claims And Evidence:**

Yes,  the task is meaningful and the claims is convincing.

**Essential References Not Discussed:**

Yes

**Experimental Designs Or Analyses:**

Yes, the experimental designs are reasonable.

**Methods And Evaluation Criteria:**

Yes, the methods and evaluation can effectively verify the claims.

**Other Comments Or Suggestions:**

None

**Other Strengths And Weaknesses:**

Strengths:
The paper is well-written overall, with technical points and experiments clearly articulated.
This task is meaningful and interesting.
The framework is feasible for this task, and its performance surpasses previous methods, according to the outcomes in the paper.

Weaknesses:
Analyzing the rationality of constraint design and the strength of constraints is crucial for exploring this new task. Conducting more experiments can contribute to this point.

Question:
Why are the constraints designed according to the content in Figure 01, and what are the reasons for such a design?
How does the proposed method perform under different constraints? If the constraints are very weak, can the direct learning still achieve good results? If the constraints are very strong, will the proposed method also be unable to handle them? Is the proposed method only effective within a relatively narrow range of constraints?

**Questions For Authors:**

None

**Relation To Broader Scientific Literature:**

This paper is related to constrained RL.

**Theoretical Claims:**

Yes, the building process of surrogate dataset by MPC is correct.

---

> ### Author Rebuttal · Authors · 2025-04-01
>
> We appreciate the reviewer’s time and effort in reviewing our work and raising valuable questions. We provide our point-by-point response as follows.
>
> > Analyzing the rationality of constraint design and the strength of constraints is crucial for exploring this new task. Conducting more experiments can contribute to this point.
>
> Thanks for the suggestion. We would like to emphasize that the constraint design follows the standard ACRL literature [1-3], where similar constraints have been considered. Specifically, box and L2 constraints directly limit the numerical range of actions, while state-dependent constraints regulate the agent's power consumption. We provide results on how DTWIL and baseline methods perform under different constraint strengths in the later sections.
>
> [1] Janaka Brahmanage, Jiajing Ling, and Akshat Kumar, “FlowPG: Action-constrained policy gradient with normalizing flows,” NeurIPS 2023.
>
> [2] Kazumi Kasaura, Shuwa Miura, Tadashi Kozuno, Ryo Yonetani, Kenta Hoshino, and Yohei Hosoe, “Benchmarking actor-critic deep einforcement learning algorithms for robotics control with action constraints,” Robotics and Automation Letters, 2023.
>
> [3] Jyun-Li Lin,Wei Hung, Shang-Hsuan Yang, Ping-Chun Hsieh, and Xi Liu, “Escaping from zero gradient: Revisiting action-constrained reinforcement learning via Frank-Wolfe policy optimization,” UAI 2021.
>
> > Why are the constraints designed according to the content in Figure 01, and what are the reasons for such a design?
>
> We believe you are asking about the constraint used in the Maze2d toy example in Figure 1. In this case, we use a box constraint, limiting action values within [−0.5, 0.5]. Our goal is to illustrate, in a simple environment with a basic constraint, how standard IL algorithms may cause the learner to deviate from the intended trajectory under action constraints, leading to task failure. We did not delve too deeply into this example, as it primarily serves as an intuitive illustration to highlight the challenges of action-constrained imitation learning.
>
> > How does the proposed method perform under different constraints? If the constraints are very weak, can the direct learning still achieve good results? If the constraints are very strong, will the proposed method also be unable to handle them? Is the proposed method only effective within a relatively narrow range of constraints?
>
> If the constraints are loose, their impact on the learner's ability to replicate the expert trajectory is minimal. Therefore, if the constraints are loose enough such that they have no effect on reproducing the expert trajectory, direct learning can still achieve good results. In our experiments, we select moderately tight constraints to ensure they significantly impact learning. Here, we show how DTWIL performs under tighter and looser constraints.
>
> | Task (Constraint Level) \ Method | CFIL-s | CFIL-sa | BCO | OPOLO | DTWIL |
> | -------- | -------- | -------- | -------- | -------- | -------- |
> | M+B (0.3)  | 0.20±0.14 | 0.21±0.03 | 0.44±0.34 | 0.44±0.16 | **0.86 ± 0.04** |
> | M+B (0.1) - paper  | 0.22 ± 0 | 0.23 ± 0.21 | 0.14 ± 0.05 | 0.20 ± 0.06 | **0.77 ± 0.04** |
> | M+B (0.07)    | 0.05 ± 0.02 | 0.06 ± 0.01 | 0.26 ± 0.10 | 0.34 ± 0.13 | **0.68 ± 0.01** |
> |||||||
> | HC+O (15) | 2065 ± 714 | **2796 ± 247** | -151 ± 13 | 47 ± 673 | **2785 ± 17** |
> | HC+O (10) - paper  | 1422 ± 1830 | 1674 ± 1316 | 6 ± 31 | -9 ± 80 | **2637 ± 26** |
> | HC+O (7)  | 54 ± 671 | 354 ± 1587 | -130 ± 9 | 99 ± 941 | **1408 ± 25** |

---

> > ### Comment · Reviewer_Fdv6 · 2025-04-02
> >
> > Thanks for the author's response. And I read the reviews of other reviewers. I think an important problem is the lack of discussions and comparisons of related work as meationed by Reviewer NcAj and NcAj. Regarding the performance comparison, the author supplemented some experiments in the rebuttal to demonstrate the effectiveness of the proposed method. Overall, the main contribution of this paper lies in using a data preprocessing method to process expert data, which I believe is helpful to this problem. However, the writing of this paper has issues of lacking discussions of relevant works. I tend to maintain my score, but I won't argue for it if others hold opposing views.

---

> > > ### Author Response · Authors · 2025-04-05
> > >
> > > We sincerely thank the reviewer for the swift response and for considering the perspectives of other reviewers.
> > >
> > > **Regarding the discussion and comparisons of related work:**
> > >
> > > The LfO methods, as the major baselines in this paper, all implicitly assume that:
> > >
> > > >(A1) The trajectories of the expert and the learner are drawn from the distributions of the same support.
> > >
> > > However, the assumption (A1) no longer holds in the Action-Constrained Imitation Learning (ACIL) setting, where the learner fundamentally cannot replicate the expert’s behavior due to the action constraints, as mentioned in Section 2 (Lines 103-119).
> > >
> > > Accordingly, DIFO [1] (mentioned by Reviewer NcAj) and the state-based imitation learning (mentioned by Reviewer G927) still rely on the assumption (A1) and encounter the same distributional mismatch issues. Therefore, we'd like to highlight that adding these comparisons does not diminish the significance of our contribution.
> > >
> > > More specifically:
> > > - DIFO [1], which belongs to the LfO category, also implicitly assumes (A1) and hence fundamentally cannot handle the action constraints. Indeed, as shown in our experimental results (please see our rebuttal for Reviewer NcAj), DIFO fails to achieve meaningful performance across all our benchmark tasks, underscoring the limitations of such an assumption in the context of ACIL.
> > > - Regarding the state-based IL methods, such as SAIL [2], they still rely on the assumption (A1), and hence these approaches encounter the same distributional mismatch issues that challenge the other LfO methods. Indeed, as validated by the experimental results (please see our rebuttal for Reviewer G927), DTWIL significantly outperforms SAIL in most of the action-constrained tasks.
> > > - Furthermore, LfO methods typically require one or multiple millions of environment interactions during training to perform well in navigation and locomotion tasks (like Hopper2d and Walker2d) [1,3,4]. In contrast, DTWIL successfully solves all the benchmark tasks using fewer than 50k interactions, demonstrating both sample efficiency and capability of handling action constraints.
> > >
> > > We believe that these comparisons do not affect the significance of our contribution; rather, they further validate the advantages of DTWIL in terms of both effectiveness and efficiency in handling action-constrained scenarios.
> > >
> > > [1] Huang et al., "Diffusion Imitation From Observation," NeurIPS 2024.
> > >
> > > [2] Liu et al., “State Alignment-based Imitation Learning,” ICLR 2020.
> > >
> > > [3] Zhu et al., “Off-Policy Imitation Learning From Observations,” NeurIPS 2020.
> > >
> > > [4] Liu et al., "Plan Your Target and Learn Your Skills: Transferable State-Only Imitation Learning via Decoupled Policy Optimization," ICML 2022.

---

### Official Review · Reviewer_NcAj · 2025-03-15

**Overall Recommendation:** 2

**Summary:**

The manuscript proposes DTWIL, a data pre-processing mechanism for translating state-action trajectories between an expert and a weaker, action-constrained learner. After synthesizing this translated data, using an MPC with a dynamic time-warping (DTW) distance measure and a forward dynamics model to provide feasible actions for a state sequence that is close to the experts, the framework encourages the use of any off-the-shelf imitation learning method.

## update after rebuttal

I appreciate the authors' answers to some of my questions; I have increased my score by one point. However, despite the author response, I still retain concerns over fair comparisons with Huang et al., 2024, which boasts strong performance on the same tasks, while using a similar formulation as what the present manuscript considers. Rather than subjecting the DiFO baseline to a simple projection layer — a strategy that the present manuscript itself argues will 'cause issues in IL' (L36;C2) such as 'severe trajectory misalignment' (L161;C2) — DiFO should be allowed to be a stronger baseline, in order to more effectively motivate the proposed approach.

**Claims And Evidence:**

L162-164;C1 — Wouldn’t it be more accurate to say that there exist trajectories within \mathcal{T} that cannot be entirely replicated by the learner, since it contains one or more actions that are outside the learner’s feasible set. For one thing, this doesn’t preclude corresponding states to be inadmissible for the learner. Another thing is that these trajectories only make up a subset of all the trajectories in \mathcal{T}. I don’t know where the intuition behind the use of the term “generally”, here, comes from.

**Essential References Not Discussed:**

Huang et al., 2024 was cited briefly, but surprisingly not used as a baseline, since it pursues the same problem and even has the same evaluation

**Experimental Designs Or Analyses:**

Section 5.1 — Old baselines; missing DIfO from Huang et al., 2024, which addresses the same problem with the same set of experiments

L192-197;C2 — The manuscript states, "... (iii) assigning a⇤t to be the first action of the selected rollout. This a⇤t would be applied to the environment to obtain the next state. This design of taking only the first action ensures that each step of the alignment process can better adapt to the remaining part of the demonstration." Common heuristic, but costly in planning complexity. What happens if more actions are executed before replanning?

**Methods And Evaluation Criteria:**

Sections 3.2, 4.2 — Why wouldn’t a diffusion policy alleviate all of these issues? The manuscript should compare with Huang et al., 2024: https://arxiv.org/pdf/2410.05429

Section 4 — Should specify what dynamics model is chosen, in the main paper content

**Other Comments Or Suggestions:**

General: The manuscript should properly use inline citations

L115;C1: ", using" —> "uses"

Pham et al., 2018b — duplicate entry

L175;C1: "describing" —> "describe"; "We start .. and then describe"

184;C2: "help a" —> "help of a"

Equation 1 — Notation error: \tao_e has not been defined as the expert trajectory yet; however, \tao was introduced as the expert state-action trajectory on L163

**Other Strengths And Weaknesses:**

N/A

**Questions For Authors:**

L116-120;C2 — Why does LfO underperform so much, if it’s trying to do the same thing? Why not compare to DIfO?

**Relation To Broader Scientific Literature:**

Huang et al., 2024 — method, idea, implementation supersedes the present manuscript

Kasaura et al., 2023; Lin et al., 2021; Brahmanage et al., 2023; Chen et al., 2024 — ACRL approaches that inspires the proposed method

Lee et al., 2021; Huang et al., 2024; Torabi et al., 2018b; Yang et al., 2019; Zhu et al., 2020; Torabi et al., 2018a — similar training objectives, problem definitions, settings

**Theoretical Claims:**

N/A — no proofs or theoretical claims

---

> ### Author Rebuttal · Authors · 2025-04-01
>
> We sincerely appreciate the reviewer’s insightful comments and valuable suggestions. We provide our point-by-point response as follows.
>
> > L162-164;C1 — Wouldn’t it be more accurate to say that there exist trajectories within \mathcal{T} that cannot be entirely replicated by the learner, since it contains one or more actions that are outside the learner’s feasible set. For one thing, this doesn’t preclude corresponding states to be inadmissible for the learner. Another thing is that these trajectories only make up a subset of all the trajectories in \mathcal{T}. I don’t know where the intuition behind the use of the term “generally”, here, comes from.
>
> Thanks for the insightful comment. Our use of "generally" was intended to convey that when the action constraints are very tight, it is typically difficult for the learner to replicate the expert trajectory. However, we acknowledge that your suggested phrasing more precisely captures the idea that certain trajectories in $\mathcal{T}$ cannot be entirely replicated due to infeasible actions. We appreciate your suggestion and will revise this part for better clarity.
>
>
> > Why wouldn’t a diffusion policy alleviate all of these issues? The manuscript should compare with Huang et al., 2024: https://arxiv.org/pdf/2410.05429
> > Huang et al., 2024 was cited briefly, but surprisingly not used as a baseline, since it pursues the same problem and even has the same evaluation.
> > L116-120;C2 — Why does LfO underperform so much, if it’s trying to do the same thing? Why not compare to DIfO?
>
> LfO also aims to imitate the state trajectory, but it tries to mimic the trajectory induced by **unconstrained actions**—trajectories that the constrained imitator cannot perfectly replicate. This limitation is precisely why we generate surrogate expert demonstrations before applying imitation learning, ensuring that the learner has feasible trajectories to follow. DIFO is also an LfO approach, and suffers from the same problem. We added a projection layer after DIFO's policy output and tested it across various action-constrained tasks.
> The results below show that across all tasks, DIFO performs poorly and fails to achieve competitive results.
>
> | Method \ Task Name | M+B | M+O | HC+B | HC+O | H+B | H+M | W+B | W+L2 |
> | -------- | -------- | -------- | -------- | -------- |  -------- | -------- | -------- |  -------- |
> | **DIFO**               | 0.27 ± 0.04 | 0.45 ± 0.09 | 8 ± 60 | -87 ± 55 | 331 ± 29  | 333 ± 32 | 6 ± 1 | 7 ± 1 |
> | **DTWIL**  | **0.77 ± 0.04** | **0.87 ± 0.04** | **2669 ± 4** | **2637 ± 26** | **2844 ± 57** | **2873 ± 240** | **61 ± 3** | **70 ± 4** |
>
>
> > Section 4 — Should specify what dynamics model is chosen, in the main paper content
>
> Thank you for the suggestion. We will move the relevant paragraph from A.2 to the main content.
>
> > L192-197;C2 — The manuscript states, "... (iii) assigning a⇤t to be the first action of the selected rollout. This a⇤t would be applied to the environment to obtain the next state. This design of taking only the first action ensures that each step of the alignment process can better adapt to the remaining part of the demonstration." Common heuristic, but costly in planning complexity. What happens if more actions are executed before replanning?
>
> We appreciate the reviewer's suggestion. This approach can indeed improve time complexity. We conducted an ablation study on varying the number of steps executed before replanning. The results show that when MPC executes multiple steps per planning cycle, the alignment performance deteriorates as the step count increases. In the Hopper environment, poor alignment further increases the risk of falling.
>
> | Steps \ Task Name | HC+B | HC+O | H+B | H+M |
> | -------- | -------- | -------- | -------- | -------- |
> | 1 step     | **2669 ± 4** | **2637 ± 26** | **2844 ± 57** | **2873 ± 240** |
> | 3 steps     | 2427 ± 0 | 2324 ± 28 | 810 ± 21 | 753 ± 43 |
> | 5 steps     | 1270 ± 464 | 2123 ± 6 | 579 ± 2 | 579 ± 10 |
>
>
> > General: The manuscript should properly use inline citations
>
> Thank you for the suggestion. We will carefully check the inline citations for the final version.
>
>
> > L115;C1: ", using" —> "uses"
> > Pham et al., 2018b — duplicate entry
> > L175;C1: "describing" —> "describe"; "We start .. and then describe"
> > 184;C2: "help a" —> "help of a"
> > Equation 1 — Notation error: \tao_e has not been defined as the expert trajectory yet; however, \tao was introduced as the expert state-action trajectory on L163
>
> Thank you for catching these typos. We will fix them in the final version.

---

### Decision · Program_Chairs · 2025-05-01

**Decision:**

Accept (poster)

**Comment:**

This paper introduces Action-Constrained Imitation Learning (ACIL), where an imitator learns from an expert but operates under stricter action constraints. Standard imitation learning fails due to mismatched occupancy measures between the expert and the constrained learner. To address this, the authors propose DTWIL, a method that generates a surrogate dataset aligning constrained actions with expert trajectories using Model Predictive Control (MPC) and Dynamic Time Warping (DTW) distance. Experiments demonstrate that DTWIL improves performance across multiple robot control tasks, outperforming existing imitation learning methods in terms of sample efficiency.

The paper introduces a new problem setting relevant for safe learning, and a novel approach, evaluated with reasonable and interesting experimental settings.

Incomplete evaluation against the most pertinent baselines is perhaps the main concern. In the AC's judgment, the concerns raised in the review regarding comparison to an "imitation-from-observation" approach are largely addressed through the comparisons to DiFO and SAIL. Of these, SAIL (2019) was designed to permit imitation with different action spaces.


SAIL is an old baseline however, and there is at least one more recent and apparently better-performing approach than SAIL for "imitation learning across *mismatched* expert and student action spaces": Ma et al 2022, SMODICE: https://arxiv.org/abs/2202.02433.
One would ideally like to see both a clear discussion of this family of works in the paper, and ideally also a comparison.

The AC would also recommend discussing the literature on addressing the "imitation gap" problem, such as Weihs et al, 2024. https://proceedings.neurips.cc/paper/2021/hash/9fc664916bce863561527f06a96f5ff3-Abstract.html